# DeNVeR: Deformable Neural Vessel Representations for Unsupervised Video Vessel Segmentation

## Abstract

This paper presents **De**formable **N**eural **Ve**ssel **R**epresentations (DeNVeR), an unsupervised approach for vessel segmentation in X-ray angiography videos without annotated ground truth. DeNVeR utilizes optical flow and layer separation techniques, enhancing segmentation accuracy and adaptability through test-time training. Key contributions include a novel layer separation bootstrapping technique, a parallel vessel motion loss, and the integration of Eulerian motion fields for modeling complex vessel dynamics. A significant component of this research is the introduction of the XACV dataset, the first X-ray angiography coronary video dataset with high-quality, manually labeled segmentation ground truth. Extensive evaluations on both XACV and CADICA datasets demonstrate that DeNVeR outperforms current state-of-the-art methods in vessel segmentation accuracy and generalization capability while maintaining temporal coherency. This work advances medical imaging by providing a robust, data-efficient tool for vessel segmentation. It sets a new standard for video-based vessel segmentation research, offering greater flexibility and potential for clinical applications.

## 1 Introduction

Coronary arteries (CAs) are essential for delivering oxygen-rich blood to the heart muscle (Dodge Jr et al., 1992). To assess coronary artery circulation and diagnose disease, cardiologists use hemodynamic measures like fractional flow reserve (FFR) and instantaneous wave-free ratio to determine the severity of stenosis (Götberg et al., 2017). Since traditional pressure wire (PW)-based techniques are invasive and involve higher risks (Stables et al., 2022), cardiologists often assess stenosis severity by

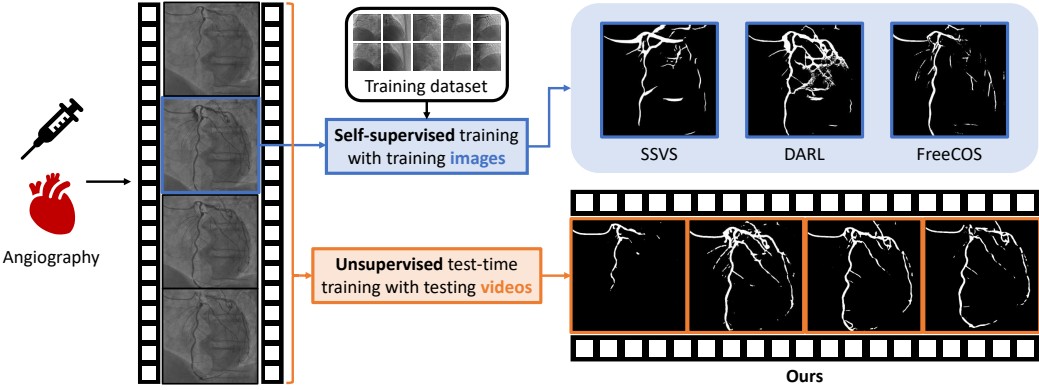

Figure 1: **Vessel segmentation method comparison.** Unlike SSVS (Ma et al., 2021), DARL (Kim et al., 2023), and FreeCOS (Shi et al., 2023), which require extensive training data, which limits their ability to generalize to new data, our method uses *unsupervised test-time training* on *testing videos*. This approach achieves superior accuracy with finer, more consistent vessel contours, demonstrating robust generalization with minimal data.

visually inspecting X-ray angiography (XRA) images. By injecting contrast agents into the coronary vessels and capturing the flow in the vessel structure on video, X-ray coronary angiography (XCA) is a common medical imaging method that exposes patients to less ionizing radiation and provides clear boundaries of the coronary arteries.

Accurate vessel segmentation remains challenging due to XCA's inherent limitations, which can obscure the severity of stenosis (Toth et al., 2014). These limitations include low signal-to-noise ratios, minimal radiation contrast (Felfelian et al., 2016), and interference from surrounding structures like catheters and bones (Maglaveras et al., 2001). The complexity of interpreting 2D projections of 3D vessels further complicates the task. Existing automatic angiographic vessel segmentation algorithms have several drawbacks. They often require professional user input and supervision to identify corresponding vessels or features in all input images (Iyer et al., 2023). The time-consuming annotation and knowledge-based training processes make it challenging to adopt these methods in practical settings. Models that take a single image as input discard critical information from the original XCA video and show reduced adaptability and compatibility when the imaging system changes or is unknown. In addition to the characteristics of X-ray angiography, involuntary organ motions and overlapping structures contribute to an increased ratio of ghosting artifacts (Liu et al., 2020a; Lin & Ching, 2005). These supervision, generalization, and dynamics issues significantly limit the application of automatic angiographic segmentation algorithms (Figure 1).

To address these challenges, we introduce DeNVeR (Deformable Neural Vessel Representations), an unsupervised approach for segmenting cardiac vessels in X-ray videos. Inspired by Deformable Sprites and using optical flow, DeNVeR starts with traditional Hessian-based filters to establish initial vessel masks as priors. It then uses a layered separation process to decompose the foreground vessel and background layers. Acknowledging the limitations of frame-by-frame processing, we enhance foreground-background segmentation through test-time optimization. This optimization incorporates neural representations of the Eulerian motion field (Holynski et al., 2021) and introduces a novel parallel vessel motion loss, thereby improving segmentation fidelity. Our approach emphasizes dynamic adaptation to cardiac movements and vessel flow, ensuring detailed, temporally consistent, and unsupervised vessel segmentation in X-ray videos. Our experimental results show significant performance improvements in predicting vessel regions compared to state-of-the-art models. The main contributions are:

- DeNVeR uses unsupervised learning on X-ray video data, leveraging the full temporal information of the videos and eliminating the need for annotated training datasets.

- Using optical flow and a unique layer separation strategy, DeNVeR enhances segmentation accuracy and adjusts during test time, improving adaptability and ensuring consistent results across cardiac conditions.

- We collect the first X-ray angiography coronary video dataset (XACV) with high-quality, manually labeled segmentation ground truth, serving as a new standard for training and evaluating video vessel segmentation models, making full use of video temporal information.

## 2 RELATED WORK

**Traditional segmentation methods.** Traditional object segmentation (Khan et al., 2020; Memari et al., 2019) requires heuristic human design rules or filters. Several methods are proposed, one of which designs the Hessian-based filter (Frangi et al., 1998) to enhance vessel filtering. Khan *et al.* (Khan et al., 2020) design retinal image denoising and enhancement of B-COSFIRE filters to perform segmentation. Memari *et al.* (Memari et al., 2019) used contrast-limited adaptive histogram equalization and designed filters to achieve the task. Another line of work proposed optimally oriented flux (OOF) (Wang & Chung, 2020; Law & Chung, 2008), which performs better in adjacent curvilinear object segmentation. These human strategy design methods do not need any training, which gives them the advantage of fast segmentation of new data. However, these methods are often confined to certain datasets and loss of generalize ability.

**Supervised and self-supervised segmentation.** In the domain of the supervised segmentation method (Soomro et al., 2019), Esfahani *et al.* (Nasr-Esfahani et al., 2016) design a Top-Hat transformation and Convolutional Neural Networks (CNNs) for segmentation. Khowaja *et al.* (Khowaja et al.,

2019) applied bidirectional histogram equalization. Another work (Yang et al., 2018; Revaud et al., 2016) utilizes image masking to reduce artifacts, which relies on paired mask datasets. The most popular vessel segmentation backbone recently is U-Net (Ronneberger et al., 2015). These methods (Fan et al., 2019; Soomro et al., 2019; Yang et al., 2019a) also require extensive and time-consuming human annotation, further limiting the application. Consequently, self-supervised learning methods are designed to elevate performance with large-scale unsupervised data. Some self-supervised learning research focuses on image painting (Pathak et al., 2016), image colorization (Larsson et al., 2017), and others (Doersch et al., 2015; Noroozi & Favaro, 2016; Ledig et al., 2017; Ren & Lee, 2018; Misra et al., 2016; Xu et al., 2019; Benaim et al., 2020; Doersch et al., 2015; Pathak et al., 2016; Gidaris et al., 2018; Misra & Maaten, 2020; Ma et al., 2021; Xie et al., 2021; Bar et al., 2022; Park et al., 2020; Wu et al., 2021; Wang et al., 2022; Alonso et al., 2021; Zhong et al., 2021). Ma *et al.* (Ma et al., 2021) and Kim *et al.* (Kim et al., 2023) proposed vessel segmentation methods with adversarial learning. Unlike these supervised and self-supervised methods requiring extensive annotations, our DeNVeR uses an unsupervised approach, training directly on test videos. It leverages optical flow and layer separation, enhancing accuracy and adaptability through test-time training. DeNVeR also utilizes temporal information, producing more coherent results than single-frame methods.

**Unsupervised segmentation methods.** Unsupervised segmentation methods fall into two categories: clustering-based and adversarial. Clustering-based approaches (Ji et al., 2019; Li et al., 2021; Do et al., 2021) like Invariant Information Clustering (IIC) by Xu et al. (Ji et al., 2019) inputs into clusters but struggle with curvilinear objects. Adversarial methods (Chen et al., 2019; Abdal et al., 2021), exemplified by Redo (Abdal et al., 2021), generate object masks by guiding generators with inputs to redraw objects in new colors.

**Video segmentation methods.** Coronary artery segmentation based on sequential images such as SVS-Net (Hao et al., 2020) use an encoder-decoder deep network architecture that utilizes multiple contextual frames of 2D and sequential images to segment 2D vessel masks. However, these supervised methods often suffer from domain gaps between datasets and cannot generalize well. Our work advances video decomposition into layers, originally proposed by Wang & Adelson (Wang & Adelson, 1994) in the 1990s, by incorporating neural network techniques (Liu et al., 2020b; 2021). Unlike traditional methods (Black & Anandan, 1991; Jojic & Frey, 2001; Ost et al., 2021; Shi & Malik, 1998; Brox & Malik, 2010), we operate unsupervised, learning deformable canonical layers to model vessel motion more effectively. Additionally, while previous research such as Yang et al. (2019b) and Ye et al. (2022) focused on general unsupervised video segmentation, we extend these concepts to the specific domain of vessel segmentation. We address motion segmentation by associating pixels with Eulerian motion (Holynski et al., 2021) clusters, adapting and extending this approach to unsupervised video vessel segmentation. Our method, DeNVeR, separates X-ray video into canonical foreground and background, per-frame masks, and dynamic transformations for realistic vessel motion representation optimized through specific loss functions.

## 3 METHOD

We propose an unsupervised algorithm for Cardiac Vessel Segmentation in X-ray videos using optical flow and test-time optimization. Our approach involves: coarse vascular region extraction (Section 3.1), layer separation using implicit neural representation (Section 3.2), background flow estimation and foreground optimization (Section 3.2), and application of specific loss and regularization terms (Section 3.4). This method addresses temporal consistency issues and enables segmentation without training data.

### 3.1 PREPROCESSING

Segmenting vascular regions from X-ray images through unsupervised methods is a highly challenging task. To facilitate our subsequent work, we employ a Hessian-based filter (Frangi et al., 1998) to generate a set of binary masks that crudely represent blood vessel regions. Specifically, our approach comprises the following two steps:

(1) Apply a Hessian-based filter to the entire sequence. The output pixel intensities range from 0 to 255, with higher numerical values indicating a stronger presence of tubular structural features.

(2) Based on the outputs from step (1), calculate the overall intensities of the entire image and set an

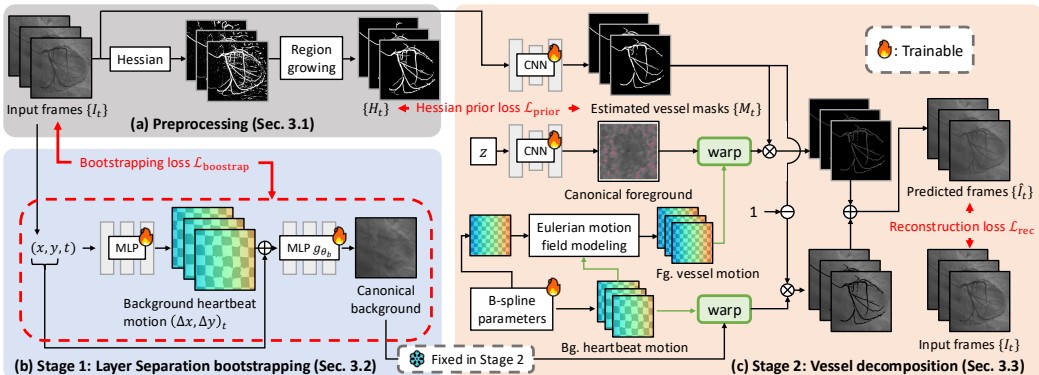

Figure 2: **Pipeline for unsupervised vessel segmentation from X-ray videos** (a) Preprocessing: Hessian-based technique with region growing for initial segmentation. (b) Stage 1: MLPs model background deformation and canonical image using bootstrapping loss. (c) Stage 2: Refine foreground vessel image, masks, and motions using B-spline parameters and warping. Reconstruction loss ensures fidelity to input frames. The pipeline trains directly on test videos without ground truth masks.

appropriate threshold to convert them into binary images. We use Otsu's method (Otsu et al., 1975) to automatically determine the threshold. The purpose of selecting the threshold is to ensure that images with higher intensities correspond to larger vascular areas. Finally, we employ a region-growing post-processing technique to eliminate noise.

Additionally, we use a pre-trained optical flow model, RAFT (Teed & Deng, 2020), to generate the initial optical flow between consecutive frames. We illustrate the preprocessing steps in Figure 2 (a).

## 3.2 LAYER SEPARATION BOOTSTRAPPING

While utilizing the Hessian-based filter allows us to quickly acquire a set of rough masks, the periodic heartbeat introduces temporal inconsistency, resulting in variations in the position of vascular regions over time. In addressing this challenge, we implement a solution by separating the input frames into foreground and background. Inspired by NIR (Nam et al., 2022), we embrace the approach of employing MLPs to learn implicit neural representations of images (Figure 2 (b)). The primary objective of each MLP is to minimize the following losses:

$$\mathcal{L}_{\text{recons}} = \sum_{x,y,t} \|\hat{I}(x,y,t) - I(x,y,t)\|_2^2, \mathcal{L}_{\text{smooth}} = \sum_{x,y,t} \|J_{g_{\theta_b}(x,y,t)}\|_1, \mathcal{L}_{\text{limit}} = \sum_{x,y,t} \|g_{\theta_f}(x,y,t)\|_1,$$

$$\mathcal{L}_{\text{boostrap}} = \mathcal{L}_{\text{recons}} + \lambda_{\text{smooth}}\mathcal{L}_{\text{smooth}} + \lambda_{\text{limit}}\mathcal{L}_{\text{limit}},$$

$$(1)$$

where $I$ and $\hat{I}$ denote the original ground truth (i.e., input RGB frames) and the output of the first MLP, respectively. To ensure flow smoothness, we introduce a penalty term for the MLP computing the background, denoted as $g_{\theta_b}$. Here, $J_{g_{\theta_b}}(x,y,t)$ represents a Jacobian matrix comprising gradients of $g_{\theta_b}$. Finally, since the stationary background should occupy the vast majority of the scene, we introduce an additional penalty term for $g_{\theta_f}$ which learns to represent the scene beyond the background. $\lambda_{\text{smooth}}$ and $\lambda_{\text{limit}}$ are weight hyperparameters. While $g_{\theta_f}$ and $g_{\theta_b}$ estimate foreground and background, they lack the spatial resolution needed for precise segmentation. Our full pipeline below refines these estimates.

## 3.3 TEST-TIME TRAINING FOR VESSEL DECOMPOSITION

Our work introduces *test-time training* as a key feature of our unsupervised segmentation method, DeNVeR. This approach adapts the model directly to test data during inference without labeled training data. Using the test video's inherent structure and patterns, the model refines its parameters, allowing it to tailor its learning to each video's unique characteristics. This capability is particularly valuable in medical imaging, where patient variability is high and personalized diagnostics are crucial. Test-time training allows DeNVeR to adjust dynamically to new, unseen cases, significantly improving diagnostic accuracy and effectiveness.

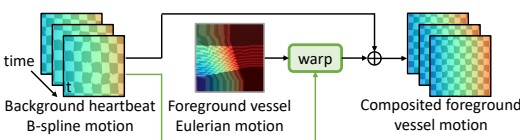

Figure 3: **Eulerian motion field modeling.** Background heartbeat uses a low-degree B-spline; foreground vessel flow uses a stationary Eulerian field. Final vessel flow combines warped Eulerian motion with background flow, capturing both factors observed in X-ray videos.

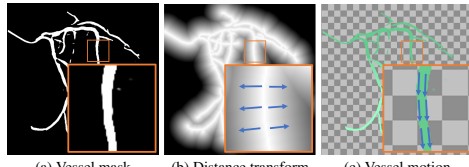

(a) Vessel mask    (b) Distance transform    (c) Vessel motion

Figure 4: **Parallel vessel motion loss.** Aligns flow direction with vessel mask direction. Uses skeletonization and distance transform to determine gradient directions. Predicted vessel motion should be perpendicular to these gradients (blue arrows).

After obtaining the Hessian-based approach as prior and bootstrapped static background, we focus on utilizing a pre-trained optical flow estimator, RAFT (Teed & Deng, 2020), to further separate the vessel and background layer and obtain vessel segmentation masks. As shown in Figure 2 (c), for each image, we use a CNN model to predict masks for the foreground and background. Both foreground and background have their canonical images. It's important to note that these images do not correspond to a specific cardiac phase. Instead, it is learned to represent the overall vessel structure across the cardiac cycle. To simplify the problem, we first compute the canonical image for the background in Stage 1 (Section 3.2) and keep it fixed. Then, in Stage 2, we optimize the canonical foreground using a CNN model with a fixed latent code $z$ (Ulyanov et al., 2018). The latent code $z$ is randomly initialized and fixed during optimization. Its purpose is to provide a consistent input for generating the canonical foreground across different optimizations. The CNN is trained during the test-time optimization process for each video. Next, we use motion flow to reconstruct respective images from the canonical images. To enhance the coherence of each frame's mask, we calculate the flow warp loss, requiring both foreground and background motion fields. Thus, we utilize the spatial and temporal B-spline to model the entire motion trajectory.

**Background motion fields.**    In the case of cardiac X-ray imaging, the background usually includes the heart and ribs, which don't experience significant displacement. Therefore, we use a B-spline with lower degrees of freedom to estimate the motion flow.

**Foreground motion fields.**    As for the foreground, we observe that the contrast agent flows out from the catheter. Therefore, we consider the Eulerian motion field, as shown in Figure 3, to be a more reasonable specification of blood flow behavior compared to the traditional motion field.

### 3.4 Losses and Regularizations

**Hessian prior loss.**    After obtaining the initial mask from preprocess (Section 3.1), we utilize a CNN model for the initial segmentation task. In this step, we aim for the model's predicted mask to closely resemble the mask generated by traditional algorithms. To achieve this, we employ a Hessian prior loss:

$$\mathcal{L}_{\text{prior}} = \sum_x H_t(x) \cdot M_t(x) + \alpha \cdot (1 - H_t(x)) \cdot (1 - M_t(x)), \tag{2}$$

where $H_t$ represents the mask of frame t generated by the preprocessing part, $M_t$ denotes the background mask of frame t predicted by the mask model, and $\alpha$ represents the foreground weight. Note that masks generated by this per-frame operation are not temporally continuous, which means that there can be sudden changes in mask predictions between adjacent frames, and our method aims to ensure smoother transitions and consistency in vessel structure across consecutive frames. Therefore, we will optimize continuity through subsequent methods.

**Parallel loss.**    Clearly, the direction of blood flow should align with the course of blood vessels (Figure 4). Hence, we design the parallel loss to achieve a parallel alignment between them. Initially, we conduct skeletonization and distance transform on the masks obtained from Section 3.1, and

calculate pixel-wise cosine similarity between these transformed masks and the predicted flow:

$$\mathcal{L}_{\text{parallel}} = \sum_x \frac{\left|\mathcal{V}(x) \cdot \hat{F}(x)\right|}{\|\mathcal{V}(x)\| \cdot \|\hat{F}(x)\|}, \mathcal{V}(x) = (\nabla_u D(x), \nabla_v D(x)), \tag{3}$$

where $D(x)$ represents the value obtained from the distance transform at pixel coordinate $x$, $\nabla_u$ and $\nabla_v$ are the image gradients from the two spatial directions, and $\hat{F}(x)$ denotes the predicted flow value at position $x$.

**Flow warp loss.** To maintain consistency in the predicted flow for both the foreground vessel and background, we introduce the flow warp loss:

$$\mathcal{L}_{\text{warp}} = \sum_{\ell \in [f,b],x} M_t^\ell(x) \cdot \frac{\|\hat{F}_t^\ell(x) - \hat{F}_{t+1}^\ell(F_{t \to t+1}(x))\|}{s_t^\ell + s_{t+1}^\ell}, \tag{4}$$

where $\hat{F}_t^\ell$ is the predicted flow at time $t$, $M_t$ represents the mask for frame t, $F_{t \to t+1}$ denotes RAFT optical flow computed from frame at time $t$ to $t + 1$, $\ell$ denotes the background layer or foreground layer, and $s_t^\ell$ is the scale of $\hat{F}_t^\ell$. The flow warp loss encourages the flow between nearby frames of both foreground vessel and background layers to follow the guidance from flow predicted by RAFT.

**Mask consistency loss.** Our current method processes a short video clip that lasts only about three seconds. Thus, we suppose the topology of the vessels remains the same during this short time period. For the predicted masks, we compare the mask at time $t$ with the deformed mask at time $t + 1$ to ensure consistency across frames. We introduce the mask consistency loss $\mathcal{L}_{\text{mask}}$:

$$\mathcal{L}_{\text{mask}} = \sum_x \left| M_t^f(x) - M_{t+1}^f(F_{t \to t+1}(x)) \right| + \left| M_t^b(x) - M_{t+1}^b(F_{t \to t+1}(x)) \right|. \tag{5}$$

**Reconstruction loss.** We use the L1 distance between the predicted image and the original image for Reconstruction loss calculation:

$$\mathcal{L}_{\text{rec}} = \left\| \hat{I}_t - I_t \right\|_1. \tag{6}$$

Our final loss function is applied to train all components shown as trainable in Figure 2 (c):

$$\mathcal{L}_{\text{final}} = \lambda_{\text{prior}}\mathcal{L}_{\text{prior}} + \lambda_{\text{parallel}}\mathcal{L}_{\text{parallel}} + \lambda_{\text{warp}}\mathcal{L}_{\text{warp}} + \lambda_{\text{mask}}\mathcal{L}_{\text{mask}} + \lambda_{\text{rec}}\mathcal{L}_{\text{rec}}. \tag{7}$$

## 4 EXPERIMENTS

### 4.1 XACV DATASET

We collect 111 complete records of coronary artery X-ray videos from 59 patients, encompassing the injection, flow through the blood vessels around the heart, and dissipation of the contrast agent. Subsequently, we establish the XACV (X-ray Angiography Coronary Video) dataset. Each video consists of an average of 86 frames of high-resolution $512 \times 512$ coronary artery X-ray images, with an equal distribution of left and right coronary arteries. We invite experienced radiologists to annotate the vascular regions, focusing on one or two frames where the contrast agent is most prominent in each video. These annotations are used only for evaluation in our method, not for training, maintaining the unsupervised nature of our approach. The data collection protocol involves several key steps, including patient preparation with informed consent and metal object removal, image capture using a Philips Allura Xper FD20 machine for standardized frontal (PA) and lateral views, DICOM file storage, and de-identification for patient privacy. Experienced radiologists perform diagnostic annotations using standardized tools and methods, with multiple annotations to enhance accuracy. Quality control measures, secure data management, and strict adherence to ethical guidelines and privacy regulations are implemented throughout the process. The XCAD dataset contains only a single image, and the CADICA video dataset does not provide corresponding ground truth. Therefore, in the following experiments, we conduct all the analyses on our collected XACV

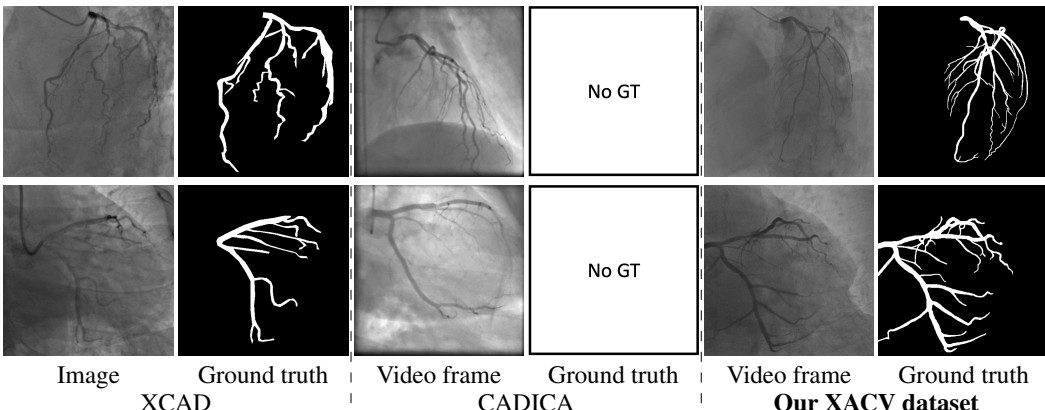

| Image | Ground truth | Video frame | Ground truth | Video frame | Ground truth |
| XCAD | | CADICA | | **Our XACV dataset** | |

Figure 5: **Comparisons between XCAD (Ma et al., 2021), CADICA (Jiménez-Partinen et al., 2024), and our XACV dataset.** (*Left*) The images from XCAD with their corresponding GTs. (*Mid*) The CADICA dataset provides video frames but without corresponding ground truth. (*Right*) Our XACV dataset with GTs meticulously labeled by experienced radiologists. Our dataset not only provides GTs with greater accuracy and detail, evident in the more nuanced vessel delineations, but also features frames of superior quality, facilitating finer and more precise segmentation results.

dataset and the corresponding GT for each sequence. In Figure 5, we show that compared to other publicly available datasets, XCAD (Ma et al., 2021) and CADICA (Jiménez-Partinen et al., 2024), our dataset exhibits finer annotations in the vascular regions, providing an advantage for future related tasks. *The development and use of our dataset have been approved by our institution's IRB.* We will make the XACV dataset publicly available.

## 4.2 BASELINE METHODS AND EVALUATION METRICS

We compare DeNVeR's performance on the XACV dataset against state-of-the-art methods, including self-supervised (SSVS (Ma et al., 2021), DARL (Kim et al., 2023), FreeCOS (Shi et al., 2023)), traditional (Hessian (Frangi et al., 1998)), and supervised (U-Net (Ronneberger et al., 2015)) approaches. For a fair comparison, we apply the same thresholding and region-growing steps to all methods and optimize heuristic thresholds using Dice scores. Following (Ma et al., 2021; Kim et al., 2023; Shi et al., 2023), we use standard metrics (Jaccard Index, Dice Coefficient, accuracy, sensitivity, specificity) and advanced metrics (NSD (Reinke et al., 2024), clDice(Shit et al., 2021), AUROC (Bradley, 1997), AUPRC (Boyd et al., 2013)) for evaluation. Due to a lack of publicly available implementations, we couldn't compare with video-based vessel segmentation methods for coronary arteries.

## 4.3 IMPLEMENTATION DETAILS

In this paper, we implement the entire deep learning architecture using PyTorch (Paszke et al., 2019) and train it with Adam optimizer (Kingma & Ba, 2015) on a single NVIDIA GeForce RTX 4090 GPU. The entire testing process, including model training and inference, takes approximately 20 minutes and utilizes 18GB of RAM. In the preprocessing stage, we compute the optical flow using RAFT (Teed & Deng, 2020).

Masks obtained from preprocessing are typically discontinuous and noisy. Therefore, we utilize deep learning methods for training. To simplify the task of vessel segmentation, we divide it into two stages. In stage 1, we use MLPs to acquire the background canonical image, with $\lambda_{\text{limit}} = 0.02$ and $\lambda_{\text{smooth}} = 0.02$. In stage 2, we employ U-Net (Ronneberger et al., 2015) to predict masks, B-spline models, and foreground canonical images. Initially, we use a warm start U-Net (Ronneberger et al., 2015) network with $\mathcal{L}_{\text{prior}}$ to generate a coarse mask, with $\mathcal{L}_{\text{prior}}$ weight set to 0.5. Then, we gradually incorporate $\mathcal{L}_{\text{parallel}}$ ($\lambda_{\text{parallel}} = 0.05$), $\mathcal{L}_{\text{warp}}$ ($\lambda_{\text{warp}} = 0.1$), $\mathcal{L}_{\text{mask}}$ ($\lambda_{\text{mask}} = 0.1$), and $\mathcal{L}_{\text{rec}}$ ($\lambda_{\text{rec}} = 0.5$) to optimize DeNVeR. Specifically, our model requires 20 minutes of runtime to process a video sequence of 80 frames. However, our method provides fully automatic segmentation without manual annotations, potentially saving significant time and resources in the long term.

Table 1: **Quantitative evaluation with different methods on the XACV dataset.** Method categories: S: Supervised, T: traditional, SS: Self-supervised, U: unsupervised. Bold indicates the best performance among traditional, self-supervised, and unsupervised methods. Our unsupervised method (DeNVeR) aims to outperform existing non-supervised approaches.

| | Input | Method | clDice | NSD | Jaccard | Dice | Acc. | Sn. | Sp. |
|---|---|---|---|---|---|---|---|---|---|
| T | Image | Hessian (Frangi et al., 1998) | $0.577_{\pm 0.062}$ | $0.321_{\pm 0.066}$ | $0.415_{\pm 0.055}$ | $0.584_{\pm 0.055}$ | $0.929_{\pm 0.015}$ | $0.451_{\pm 0.062}$ | $\mathbf{0.990_{\pm 0.008}}$ |
| S | Image | U-Net (Ronneberger et al., 2015) | $0.757_{\pm 0.114}$ | $0.603_{\pm 0.126}$ | $0.638_{\pm 0.126}$ | $0.771_{\pm 0.107}$ | $0.956_{\pm 0.015}$ | $0.711_{\pm 0.151}$ | $0.986_{\pm 0.008}$ |
| SS | Image | SSVS (Ma et al., 2021) | $0.408_{\pm 0.057}$ | $0.216_{\pm 0.039}$ | $0.355_{\pm 0.046}$ | $0.522_{\pm 0.050}$ | $0.905_{\pm 0.013}$ | $0.471_{\pm 0.056}$ | $0.960_{\pm 0.009}$ |
| | Image | DARL (Kim et al., 2023) | $0.605_{\pm 0.065}$ | $0.300_{\pm 0.058}$ | $0.464_{\pm 0.064}$ | $0.631_{\pm 0.060}$ | $0.929_{\pm 0.014}$ | $0.547_{\pm 0.060}$ | $0.978_{\pm 0.014}$ |
| | Image | FreeCOS (Shi et al., 2023) | $0.639_{\pm 0.101}$ | $0.461_{\pm 0.087}$ | $0.506_{\pm 0.135}$ | $0.660_{\pm 0.131}$ | $0.941_{\pm 0.015}$ | $0.554_{\pm 0.152}$ | $0.988_{\pm 0.004}$ |
| U | Video | **DeNVeR(Ours)** | $\mathbf{0.704_{\pm 0.081}}$ | $\mathbf{0.515_{\pm 0.101}}$ | $\mathbf{0.584_{\pm 0.082}}$ | $\mathbf{0.733_{\pm 0.066}}$ | $\mathbf{0.947_{\pm 0.014}}$ | $\mathbf{0.656_{\pm 0.091}}$ | $0.985_{\pm 0.006}$ |

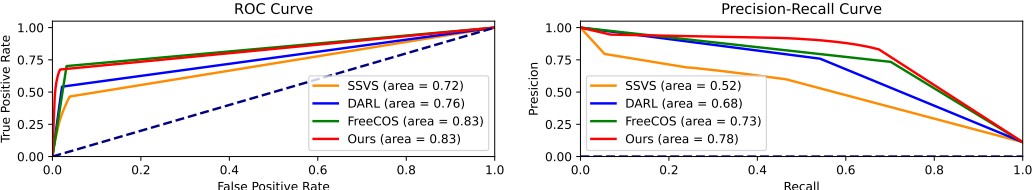

Figure 6: **AUROC and AUPRC results.** Our model performs favorably against other methods on both AUROC and AUPRC.

## 4.4 COMPARISON

Table 1 reports the performance of video vessel segmentation on the XACV dataset between our proposed DeNVeR and the baseline methods. Although our method is unsupervised, for comparison with other supervised or self-supervised methods, we still partition the entire dataset into training and testing sets in an 8:2 ratio. All the results recorded in Table 1 are obtained on the testing set.

Since supervised training and testing data are from the same dataset (in-domain setting), its performance will be better than that of self-supervised or unsupervised methods. However, it is worth noting that in this scenario, our method does not require any labels and can still outperform existing self-supervised methods. Also, we test the CADICA dataset to compare the generalization ability of supervised training and our proposed unsupervised training in Figure 8. We find that supervised methods are limited by the domain of their training data and thus struggle to generalize well. Our method, while requiring test-time training, can adapt to various datasets in an unsupervised manner. This allows for greater flexibility and generalization across different types of vascular video data.

In comparison with the traditional Hessian-based filter, our method achieves a 16.9% improvement in the Jaccard Index and a 14.9% increase in the Dice Score, indicating a significant enhancement in performance while utilizing it as a prior. While our method is more complex than supervised approaches, it eliminates the need for costly and time-consuming manual annotations. The test-time training phase, though computationally intensive, is a one-time process per video. For self-supervised methods, we follow their tutorials to augment the training dataset and generate synthetic masks for training. Each model is trained for at least 100 epochs. The results indicate that FreeCOS (Shi et al., 2023) performs the best among them, but our approach still shows a 7.7% improvement in the Jaccard Index and a 7.3% improvement in the Dice Score compared to it. It is worth noting that, due to the sensitivity of the Hessian-based approach to the chosen threshold and its greater bias, under our intentionally selected optimal conditions, the performance of SSVS may be slightly lower than that of the Hessian-based filter.

We calculate the AUROC and AUPRC in Figure 6. We normalize the model's final layer output to [0, 1] to use it as the probability for calculating AUROC and AUPRC. Our model performs favorably against other methods on both AUROC and AUPRC. We also provide visual comparison results in Figure 7, demonstrating our vessel segmentation results are more accurate, complete, and closer to the ground truth masks. Moreover, in some sequences, our method even performs on par with supervised U-Net (Ronneberger et al., 2015), as U-Net might face an overfitting problem with insufficient training data. Additionally, we provide visual comparisons on the CADICA (Jiménez-Partinen et al.,

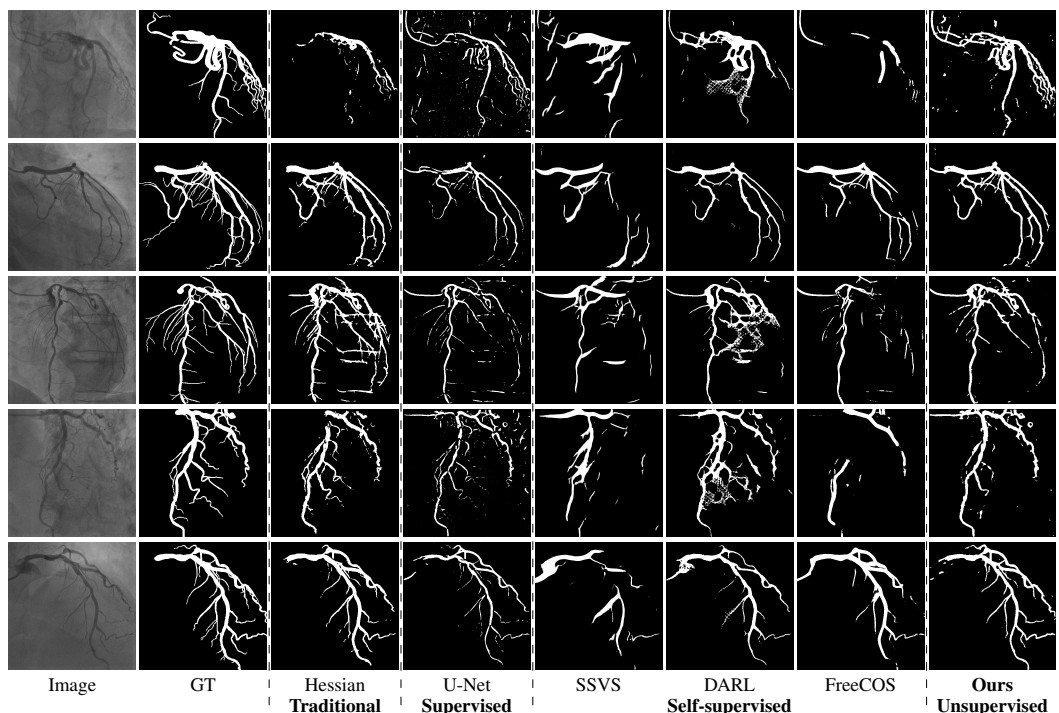

| Image | GT | Hessian | U-Net | SSVS | DARL | FreeCOS | **Ours** |
|---|---|---|---|---|---|---|---|
| | | **Traditional** | **Supervised** | | **Self-supervised** | | **Unsupervised** |

Figure 7: **Visualization results on the vessel segmentation.**

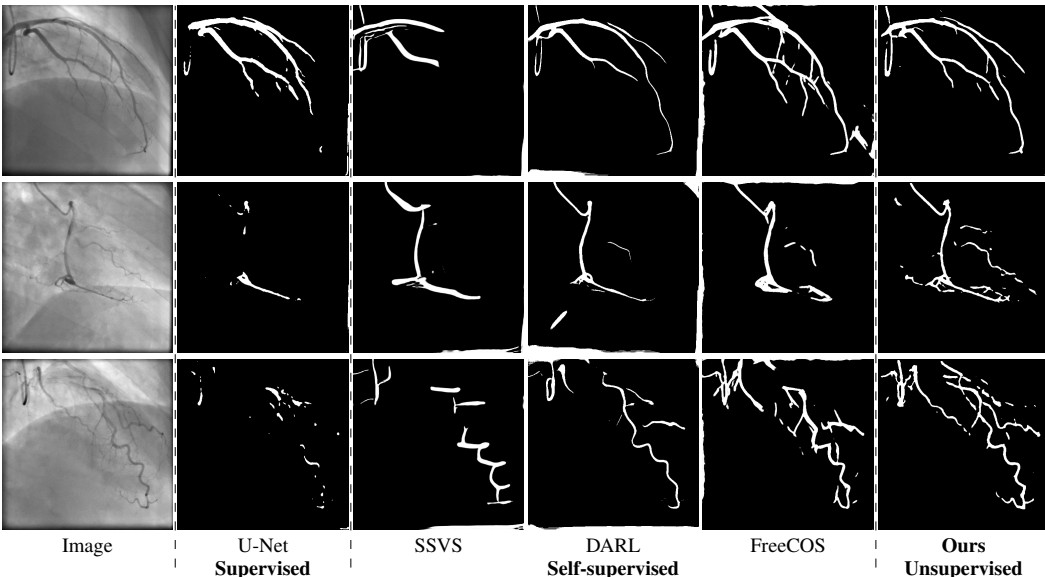

| Image | U-Net | SSVS | DARL | FreeCOS | **Ours** |
|---|---|---|---|---|---|
| | **Supervised** | | **Self-supervised** | | **Unsupervised** |

Figure 8: **Results on CADICA (Jiménez-Partinen et al., 2024) dataset.** Supervised methods cannot generalize well to a new dataset and suffer from the domain gaps between training (our XACV) and testing datasets (CADICA). Our method, although requiring test-time training, can adapt to various datasets in an unsupervised manner. The CADICA dataset does not contain GT and is the only video vessel dataset publicly available. Therefore, we can only demonstrate qualitative comparisons.

2024) dataset, which is also a coronary artery X-ray video dataset but without ground truth labeling. Figure 8 demonstrates that our test-time training scheme generalizes better than existing methods. Due to the space limit, we provide more visual comparisons in the appendix.

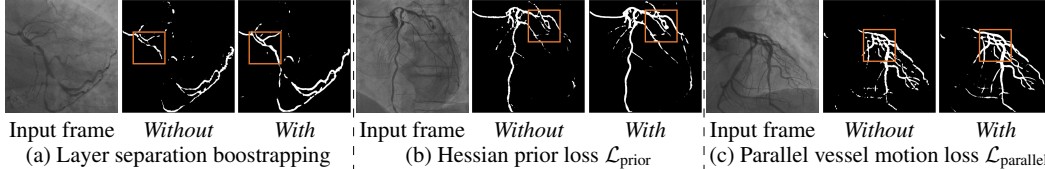

Input frame    *Without*    *With*     Input frame    *Without*    *With*     Input frame    *Without*    *With*
(a) Layer separation boostrapping     (b) Hessian prior loss $\mathcal{L}_{\text{prior}}$     (c) Parallel vessel motion loss $\mathcal{L}_{\text{parallel}}$

Figure 9: **Visual comparisons of ablation studies.**

Table 2: **Ablation study.**

| Bootstrap | $\mathcal{L}_{\text{prior}}$ | $\mathcal{L}_{\text{parallel}}$ | Jaccard | Dice | Acc. | Sn. | Sp. |
|---|---|---|---|---|---|---|---|
| - | ✓ | ✓ | 0.4821 | 0.6462 | 0.9359 | 0.5722 | 0.9814 |
| ✓ | - | ✓ | 0.4630 | 0.6333 | 0.9321 | 0.5319 | **0.9857** |
| ✓ | ✓ | - | 0.5394 | 0.6971 | 0.9428 | 0.6049 | 0.9856 |
| ✓ | ✓ | ✓ | **0.5840** | **0.7339** | **0.9479** | **0.6567** | 0.9855 |

## 4.5 ABLATION STUDY

**Layer separation bootstrapping.** To validate the effectiveness of the layer separation bootstrapping, we train foreground and background canonical images using the same representation. The results are shown in Table 2, where optimizing both foreground and background canonical images simultaneously leads to a decrease in the Dice score by 0.0877. The comparison is shown in Figure 9 (a), where the orange area indicates the difference between without and with Layer separation bootstrapping. The bottom-right corner shows a zoom-in patch, highlighting the significant effect of the bootstrapping step.

**Hessian prior loss $\mathcal{L}_{\text{prior}}$.** To test the effect of the Hessian prior loss, we remove the Hessian prior loss. As a result, the segmentation performance, as shown in Table 2, also decreases the Dice score by 0.1006. The comparison between the without and with $\mathcal{L}_{\text{prior}}$ is shown in Figure 9 (b), where the orange area indicates the difference between them. The zoom-in patch shows that our model predicts less noticeable vascular regions incorporating the Hessian prior loss $\mathcal{L}_{\text{prior}}$.

**Parallel vessel motion loss $\mathcal{L}_{\text{parallel}}$.** We conduct an experiment to assess the effect of the parallel vessel motion loss by removing it from the training pipeline. As shown in Table 2, the segmentation performance decreases the Dice score by 0.0368. Without this loss to enforce parallelism between blood and vessels, the segmentation results are negatively affected. In addition, the comparison between without and with $\mathcal{L}_{\text{parallel}}$ is shown in Figure 9 (c). The zoom-in patch shows that the image with $\mathcal{L}_{\text{parallel}}$ has clearer segmented vascular regions.

The improvements from individual components may appear marginal. However, their cumulative effect leads to overall superior performance compared to baselines. In Figure 9, we provide visual comparisons of ablation studies, demonstrating that these components are essential for clear and complete vessel segmentations. These components help connect disconnected or over-segmented vessels in specific cases.

## 5 CONCLUSIONS

This paper introduces DeNVeR, an unsupervised test-time training framework for vessel segmentation in X-ray video data. DeNVeR utilizes optical flow and layer separation techniques to accurately segment vessels without requiring annotated datasets. Quantitative and qualitative evaluations on the XACV and CADICA datasets show that DeNVeR outperforms existing image-based self-supervised methods, offering precise delineation of vessel boundaries critical for medical diagnosis and treatment.

**Limitations.** Our unsupervised method is sensitive to preprocessing filters, potentially misidentifying non-vascular structures as vessels. DeNVeR's runtime (20 minutes for 80 frames) and computational requirements are also limiting factors. Additionally, our motion-based approach does not apply to datasets without contrast agent flow, such as retinal vessel images.

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

# A APPENDIX

## A.1 ADDITIONAL VISUALIZATION RESULTS

Figure 10 and Figure 11 demonstrate a comprehensive comparison where we consider supervised learning (Isensee et al., 2018) as the upper bound for the vessel segmentation task, as well as all baseline methods mentioned in the main paper. For the supervised learning approach, both image-based and video-based inputs were considered. The image-based input utilized only the annotated image, while the video-based input involved using the annotated image along with two preceding and two subsequent frames, totaling five frames, as input. The results show that although supervised learning theoretically offers the best performance, our method achieves close to those of supervised learning methods without ground truth. Additionally, we found that using five consecutive images as input for nn-UNet (Isensee et al., 2018) were only slightly better than using a single image as input. In contrast, our method exhibits significant improvement compared to both the traditional Hessian-based filter and self-supervised methods, demonstrating that the robust performance of our approach is not solely attributed to the increase in input images. We showcase some examples at the following URL: `https://colab.research.google.com/drive/1IYGiJECwAaoLPq7KGHQE_dvtrdHz9fUA?authuser=2&hl=zh-tw#scrollTo=n1ppvOhqbRkV`

## A.2 TEMPORAL COHERENCY

Our method takes an entire X-ray video as input, thus producing segmentation results with better temporal coherency. Temporal coherency is essential for making medical diagnoses, especially when dealing with blood flow in vessels. Therefore, we conduct visual comparisons between our method and other compared methods by slicing horizontally or vertically and stacking the segmentation results. The results in Figure 12 show our method strikes a better balance between segmentation accuracy and temporal coherency. While other baseline methods either produce false segmentation results or do not maintain consistent prediction along the temporal dimension.

## A.3 IMPACT OF PRIOR

We add experiments demonstrating how the Hessian prior affects subsequent results, including ablation studies with different prior qualities. In our experiments, We replace the Hessian prior mask with a better mask (FreeCOS prediction) and observe a 2.5% improvement in dice score. We also provide visual results in Figure 13.

## A.4 MODEL AND TRAINING DETAILS

We elaborate on the architectural details and training methodologies for all neural network components.

### A.4.1 STAGE1: LAYER SEPARATION ON BOOTSTRAPPING

This MLP (Multi-Layer Perceptron) model consists of these main components:

- Input Layer: Input dimension is 3 (color channels).
- Hidden Layer 1: Takes input of dimension 3 and outputs a dimension of 2. This layer has 256 neurons, with 4 hidden layers and an outermost linear layer.
- Hidden Layer 2: Takes input of dimension 2 and outputs a dimension of 3. This layer also has 256 neurons, with 4 hidden layers and an outermost linear layer.
- Output Layer: Takes input of dimension 3 and outputs a dimension of 4. This layer has 256 neurons, with 4 hidden layers and an outermost linear layer.

Important hyperparameters:

- $\lambda_{\text{smooth}}$: Controls the weight of the smoothness term in the bootstrapping loss. We set it to 0.001.

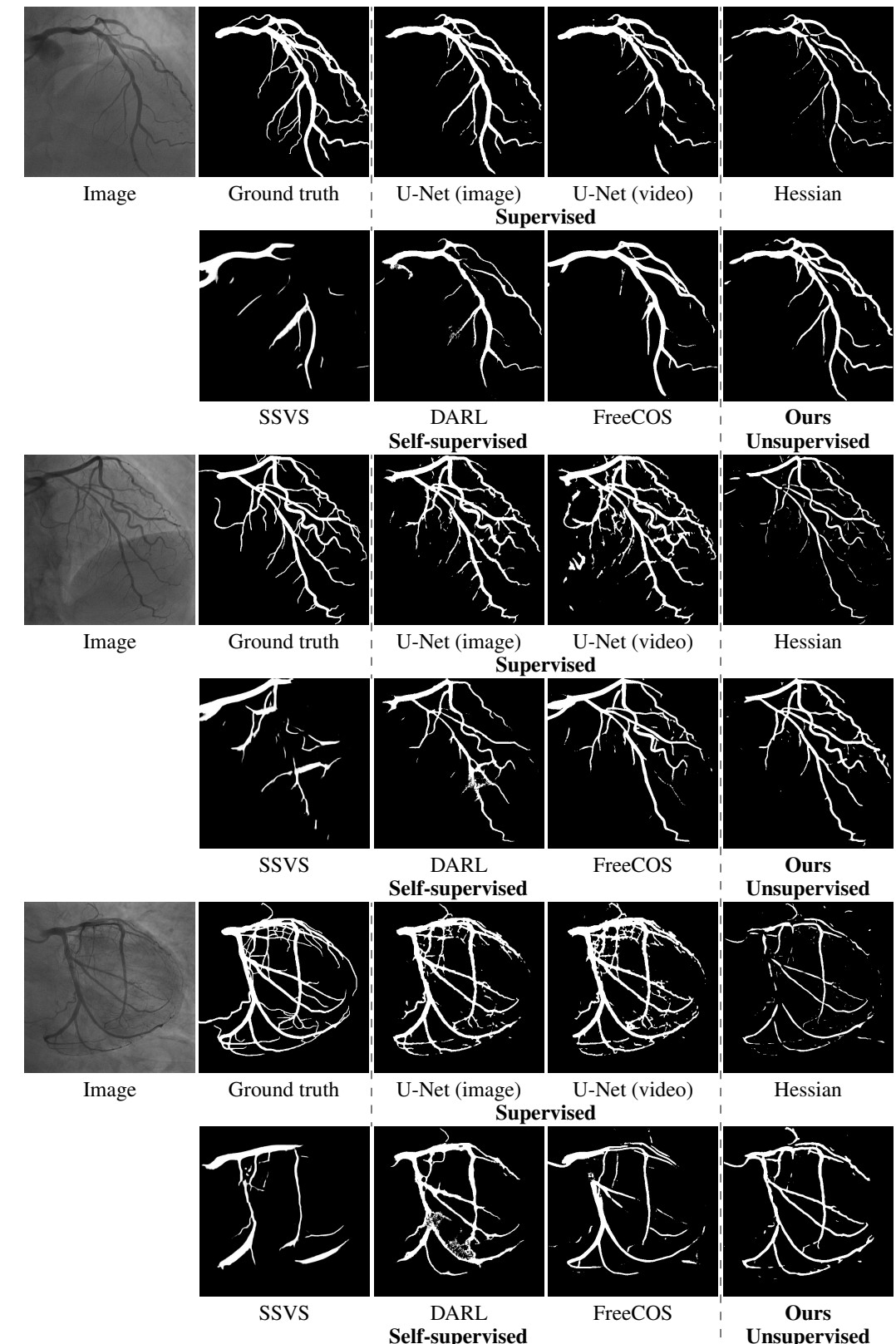

Figure 10: **Additional visualization results on the vessel segmentation.**

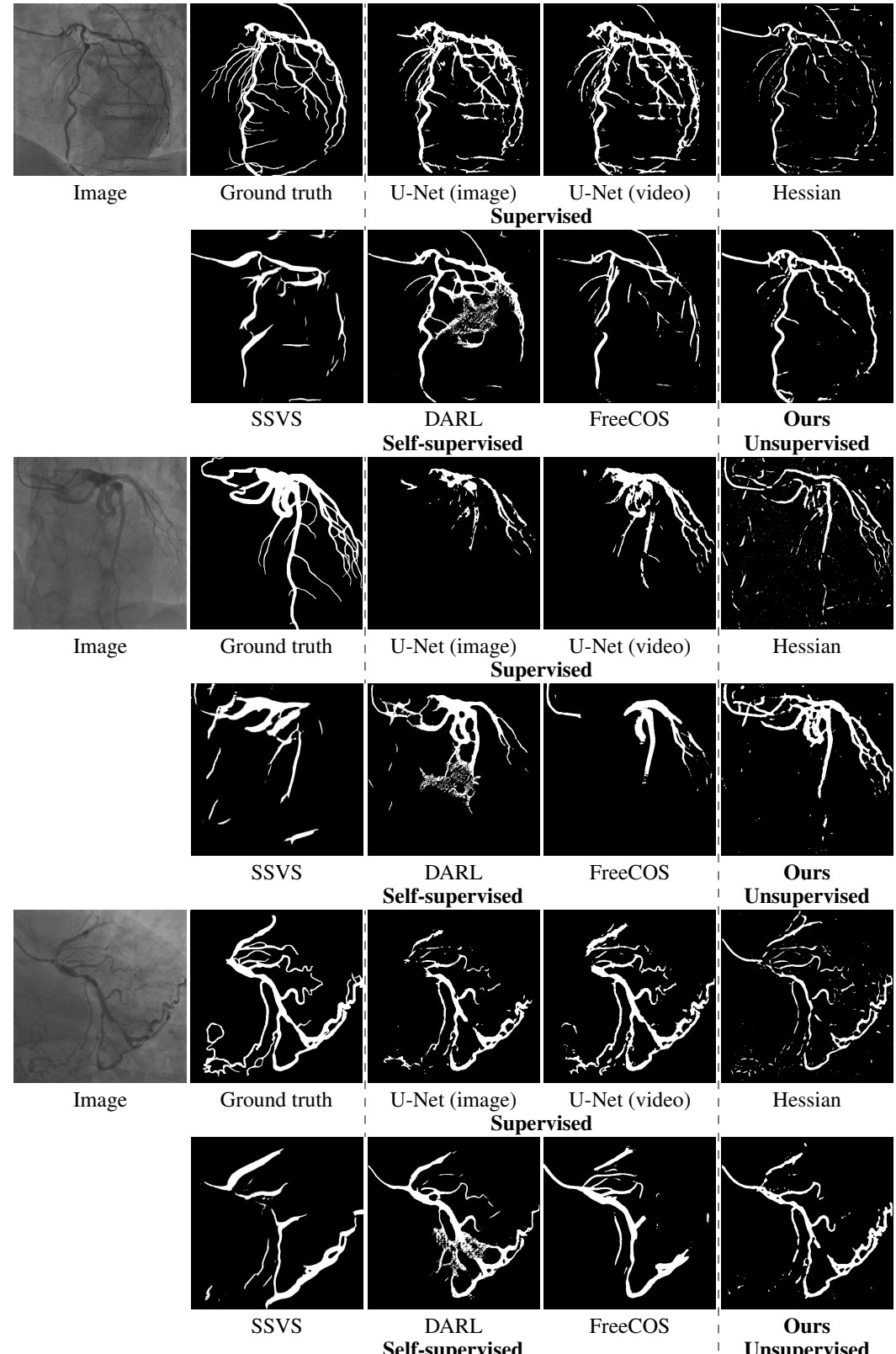

Figure 11: **Additional visualization results on the vessel segmentation.**

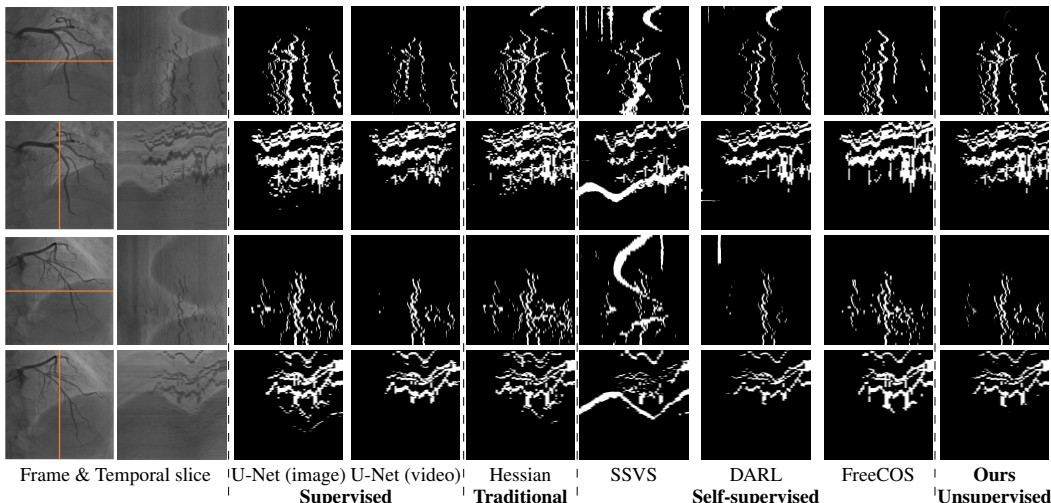

| Frame & Temporal slice | U-Net (image) | U-Net (video) | Hessian | SSVS | DARL | FreeCOS | **Ours** |
|---|---|---|---|---|---|---|---|
| | **Supervised** | | **Traditional** | | **Self-supervised** | | **Unsupervised** |

Figure 12: **Temporal coherency.**

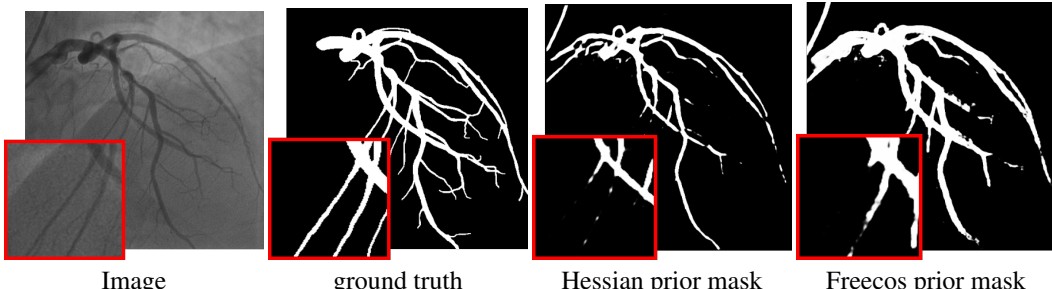

| Image | ground truth | Hessian prior mask | Freecos prior mask |
|---|---|---|---|

Figure 13: **Impact of prior.** We test the impact of the prior on our model. Replacing the original Hessian prior with the FreeCOS prediction results in a 2.5% improvement in dice score. Red zoom-in patches show that the FreeCOS-based prior has better predictive capabilities.

- $\lambda_{\text{limit}}$: Regularizes the foreground MLP in the bootstrapping loss. We set it to 0.02.

### A.4.2 STAGE 2: VESSEL DECOMPOSITION

In stage 2, We employ different standard U-Nets with three down and three up layers to predict masks and foreground canonical images. Both models utilize CNNs with 3x3 kernels, strides of 1, and padding of 1. We use batch norm and bilinear downsampling or upsampling after each layer in the U-Nets.

**Training setting.** We set the batch size to 16, including 512x512 image resolution, and trained on a 4090 GPU. Training on 80-90 cardiac images takes approximately 20 minutes.

### A.5 HYPERPARAMETER SENSITIVITY ANALYSIS

We conduct a hyperparameter sensitivity analysis in Figure 14, including the weights of various losses. Our method is robust and not sensitive to different hyperparameters.

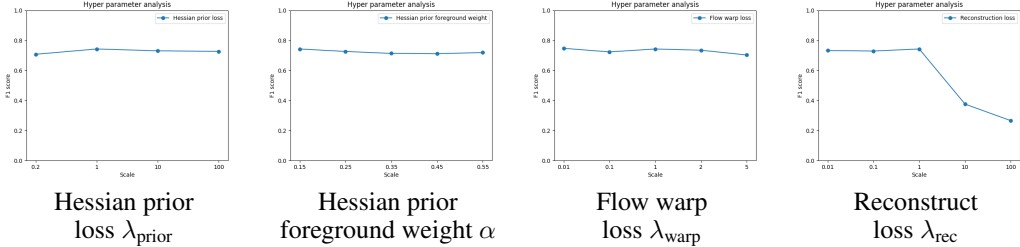

| Hessian prior loss $\lambda_{\text{prior}}$ | Hessian prior foreground weight $\alpha$ | Flow warp loss $\lambda_{\text{warp}}$ | Reconstruct loss $\lambda_{\text{rec}}$ |
|---|---|---|---|

Figure 14: **Hyperparameters sensitivity analysis.** Including weights of various losses. Our method is robust and not sensitive to different hyperparameters.

