# OpenReview forum: "DeNVeR: Deformable Neural Vessel Representations for Unsupervised Video Vessel Segmentation"
_ICLR.cc/2025/Conference — ICLR 2025 Conference Withdrawn Submission_

### Official Review · Reviewer_RUb7 · 2024-10-26

**Soundness:** 2
**Presentation:** 2
**Contribution:** 2
**Rating:** 3
**Confidence:** 5

**Summary:**

This paper introduces DeNVeR, an unsupervised approach for segmenting cardiac vessels in X-ray angiography videos without requiring annotated datasets. By leveraging temporal information in video data, DeNVeR uses optical flow and a layer separation technique to enhance segmentation accuracy and adaptability at test time, ensuring consistent performance across varied cardiac conditions. The authors creat the XACV dataset—claimed to be the first X-ray angiography coronary video dataset with high-quality, manually labeled ground truth. DeNVeR outperforms baseline methods.

**Strengths:**

DeNVeR operates without annotated training datasets, using an unsupervised learning approach that takes advantage of the complete temporal information available in X-ray video data. This enables effective vessel segmentation directly from the video sequence.

By employing optical flow analysis and an innovative layer separation strategy, DeNVeR refines segmentation results dynamically at test time, achieving decent adaptability and consistent performance across various cardiac conditions.

XACV is claimed to be the first coronary angiography video dataset with high-quality, manually labeled segmentation ground truth. XACV sets a new benchmark for training and evaluating video vessel segmentation models, fully leveraging video-based temporal data for improved segmentation fidelity.

**Weaknesses:**

Its broader implications for the ICLR community is unclear, especially how this could benefit the general computer vision and machine learning community.

The introduction of the XACV dataset is valuable, but it also highlights the niche focus of the work. It shows the research might be limited to a small community, without wider research adoption for general CV and AI.

The approach, while powerful, may be overly complex for the specific problem domain without demonstrated flexibility across different datasets or applications. To establish robustness, an evaluation of DeNVeR on broader computer vision tasks could show its adaptability.

"There is no free lunch" It is not clear what would be the limitation of the proposed method, especially without using manual annotation.

How would the clinicians know the uncertainty and trustworthiness of the results?

**Questions:**

How this would help the CV and AI community?

Is this method overfit and specific to this domain application?

What would be the limitation of the method?

---

### Official Review · Reviewer_C1Zb · 2024-11-03

**Soundness:** 3
**Presentation:** 2
**Contribution:** 3
**Rating:** 5
**Confidence:** 4

**Summary:**

The paper propose a method DeNVeR Deformable Neural Vessel Representations. The method utilizes optical flow and layer separation techniques, enhancing segmentation accuracy and adjusts during test time, improving adaptability and ensuring consistent results across cardiac conditions. And during the training , the paper leverage the full temporal information of the videos and eliminating the need for annotated training datasets.

**Strengths:**

The author have rich theory, and solid basic skills. This paper achieve the unsupervised segmentation by summarizing the many methods and designing the fitting architecture. Especially the designing of losses, there are numerous work to be finish, and the performance in the experiment seems good.

**Weaknesses:**

1、The baseline lacks the unsupervised model to compare.
2、The paper need explain the reason for guidance, such that significant of optical flow, latent code, etc.
3、The paper add one group experiment for unsupervised image segmentation not vedio to prove the effect of model in single image.
4、The paper seems like the integration of all kinds of method.

**Questions:**

The loss  function seems be written mistake, the prediction of foreground as one as possible.

---

### Official Review · Reviewer_7dwa · 2024-11-04

**Soundness:** 3
**Presentation:** 3
**Contribution:** 3
**Rating:** 6
**Confidence:** 4

**Summary:**

This paper proposed a fully unsupervised learning method for coronary vessel segmentation in X-ray videos. It achieved this by using layer separation, which takes advantage of different motion patterns in the vessel layer (foreground) and the rest structures (background) and across-frame consistency of their appearance. It also employed a test-time training method to address the high variability in medical imaging data. Overall, since unsupervised coronary vessel segmentation in X-ray videos is an underexplored field, the proposed method, showing descent performance, is a valuable contribution. In addition, this paper also contributes the first X-ray coronary angiography video dataset with fine labels, which is a valuable source for the field.

**Strengths:**

- Unsupervised coronary vessel segmentation in X-ray videos is an underexplored field, and the proposed method, showing descent performance, is a valuable contribution.
- The method is well designed and clearly presented.
- Extensive experiments and ablation analysis.

**Weaknesses:**

- Figs 7 and 8 show relatively easy scenarios for coronary vessel segmentation, where there are few interfering objects such as ribs, catheters, and surgical wires. Authors may want to show more challenging cases.
- Small vessels are not being well segmented in Figs 7 and 8, and there are also broken vessel segmentations. Where is the bottleneck? In other words, which module(s) are responsible for the false negative here?
- Authors may consider show more intermediate results (e.g. input/output of each module/step) to help readers better understand where the strengths and weaknesses of the design are.
- There is a trend of using foundation models or pre-trained large models to tackle the small-dataset supervised or unsupervised segmentation problems. I think including such a baseline is important in evaluating the contribution of this work.
- Authors may also want to report how accurate the segmentation boundary is (e.g. Harsdorf distance), as boundary accuracy is essential for downstream tasks such as FFR calculation.
- Numerous losses were weighted summed in the training. How sensitive is the model performance to the choice of weights?

**Questions:**

- In Fig. 2, why implicit neural representation (MLP) was used in stage 1 to fit the canonical background, whereas DIP was used in stage 2 to fit the canonical foreground? Why not using the same method or the other way around? What's the motivation here?
- In Fig. 3, the background motion should include both heartbeat and breathing motion. Shouldn't these two motion patterns be separated before used to warp the vessel Eulerian motion?

---

### Official Review · Reviewer_KrRM · 2024-11-04

**Soundness:** 2
**Presentation:** 2
**Contribution:** 1
**Rating:** 3
**Confidence:** 4

**Summary:**

In this submission, the authors present “Deformable Neural Vessel Representations,” a highly specialized method for vascular segmentation in X-ray angiography videos. The proposed method is an unsupervised approach that uses “a novel layer separation bootstrapping technique, a parallel vessel motion loss, and the integration of Eulerian motion fields for modeling complex vessel dynamics” (L 16-18). The method outperforms other unsupervised approaches in the segmentation task but does not outperform a simple supervised U-Net baseline.

Experiments are conducted on a single dataset named XACV, which is newly released with the submission.

**Strengths:**

- The method efficiently combines modern techniques for the complex task of vessel segmentation in videos. These include test-time training, multiple losses, and Eulerian motion fields. The authors clearly demonstrate in an ablation study how each component contributes to the overall performance gain.

- Existing unsupervised approaches are outperformed.

- The authors provide source code, which is a plus for reproducibility. However, the codebase is nested and not well-documented, making a reproducibility check challenging within a reasonable time frame, which led me not to run the code myself. Overall, this is still a positive point.

**Weaknesses:**

- In my opinion, the work at hand is not a perfect conference fit due to its heavily applied nature on a very specific topic: unsupervised vessel segmentation in X-ray videos. Submission of this nice work to a dedicated medical image analysis conference could reach an audience which is more familiar and interested in this work.

- Experimentation. The method is evaluated on a single dataset, which is also newly proposed. However, this raises a question: For this very specific task, where the XACV dataset now exists, why do we need an unsupervised method when a supervised method performs better, and annotation could be done in reasonable time?

- Topological metrics are very important for evaluating the faithfulness of vessel segmentation; I suggest adding metrics such as Betti errors to the evaluation table and discussing the results in this regard.

- Hyperparameter selection. What was the range of hyperparameters tested, and how much time or resources were used for tuning? How were the hyperparameters for the four baseline methods specifically chosen? I believe clearly describing the hyperparameter search is essential for reproducibility. For example these additional results could be presented in additional tables.

**Questions:**

Have the authors used their method to train a general representation and then fine-tuned it on the labels they have? I think this would be an interesting baseline, and if successful, it would strengthen the method, showing that pretraining helps.

---

### Note · Authors · 2024-11-13

I have read and agree with the venue's withdrawal policy on behalf of myself and my co-authors.